# Environmental, Economic and Social Impact Assessment: Study of Bridges in China’s Five Major Economic Regions

**DOI:** 10.3390/ijerph18010122

**Published:** 2020-12-26

**Authors:** ZhiWu Zhou, Julián Alcalá, Víctor Yepes

**Affiliations:** Institute of Concrete Science and Technology (ICITECH), Universitat Politècnica de València, 46022 València, Spain; jualgon@cst.upv.es (J.A.); vyepesp@cst.upv.es (V.Y.)

**Keywords:** sustainable development, LCIA, LCCA, SILA, cable-stayed bridge, GDP

## Abstract

The construction industry of all countries in the world is facing the issue of sustainable development. How to make effective and accurate decision-making on the three pillars (Environment; Economy; Social influence) is the key factor. This manuscript is based on an accurate evaluation framework and theoretical modelling. Through a comprehensive evaluation of six cable-stayed highway bridges in the entire life cycle of five provinces in China (from cradle to grave), the research shows that life cycle impact assessment (LCIA), life cycle cost assessment (LCCA), and social impact life assessment (SILA) are under the influence of multi-factor change decisions. The manuscript focused on the analysis of the natural environment over 100 years, material replacement, waste recycling, traffic density, casualty costs, community benefits and other key factors. Based on the analysis data, the close connection between high pollution levels and high cost in the maintenance stage was deeply promoted, an innovative comprehensive evaluation discrete mathematical decision-making model was established, and a reasonable interval between gross domestic product (GDP) and sustainable development was determined.

## 1. Introduction

The most common greenhouse gases in the Earth’s atmosphere include water vapour (H_2_O), carbon dioxide (CO_2_), methane (CH_4_), nitrous oxide (N_2_O), ozone (O_3_) and chlorofluorocarbons (CFC). The concentration of carbon dioxide in the atmosphere has a dominant influence on global warming [1,2]. According to predictions by the United Nations, the world’s population will reach 9.8 billion in 2050 [3]. Population shifts will result in a massive consumption of resources and a rapid growth of energy requirements [4]. This makes the sustainable development of the construction industry, which accounts for 44% of all energy consumption, become more urgent [5,6]. What is the key to sustainable development? It is to reduce environmental, economic and social impacts [7]. Thus, the scope of research is expanded to the economic and social aspects, and the close correlation between producers and consumers is increased [8].

To avoid the serious consequences brought about by climate change, efforts should be made to substantially reduce the emission of greenhouse gases. Hansen et al. revealed that the concentration of carbon dioxide in the atmosphere must be less than 350 parts per million (ppm); otherwise, climate change will get worse [9]. The analysis of the latest global atmospheric observations by the World Meteorological Organisation shows that the global mean surface mole fractions of CO_2_, CH_4_, and N_2_O reached new highs in 2015, i.e., 400.0 ± 0.1 ppm, 1845 ± 2 parts per billion (ppb), and 328.0 ± 0.1 ppb, respectively. These values constitute 144%, 256% and 121% of the pre-industrial levels (before 1750), respectively [10].

Low-carbon energy consumption and the reduction in greenhouse gas emissions from the construction industry are particularly critical [11]. Lin and Liu. cited the CO_2_ emissions from commercial and residential buildings in China, surveyed by the Index Decomposition Analysis (IDA), which concluded that emissions from the construction industry account for 30–50% of the total emissions [12]. Science researchers all over the world have proposed measures to reduce environmental pollution caused by the construction industry. For the accuracy and systematisms of the research, LCIA was introduced to solve problems facing the construction industry [13,14]. Standardised provisions for multiple systemic analysis methods were given in ISO 14040 and ISO 14044 [15].

Table 1 shows a comparative analysis of the latest research results of LCIA, LCCA and SILA.

First, this study aims to evaluate the impact of LCIA~LCCA~SILA (2L~1S) on six bridges in five different regions of China. This study will fill the gap in the research for bridges of similar structure and purpose across regions, provinces, and economic belts in this field. Secondly, the process of 2L~1S is digitised and visualised to display the research results more intuitively. Thirdly, this study also considers the mutual influence between 2L~1S and the regional economic belts, to obtain the optimal interval and scope of influence.

The main purpose of this research is to analyse and study the comprehensive impact of bridges of the same structure in different regional economic zones on the environment, economy and society (three pillars) throughout their life cycle through software. In addition, discussed the correlation between regional economic development and the three pillars through modelling.

The innovations of the research are as follows: (1) break through the usual sustainability research and only focus on textual descriptions, without accurate modelling data descriptions; (2) the selected research case represents the influence status between the main economic belts in China and has important guiding significance for the future planning of the government and related departments.

The rest of this work will be divided into the following sections: Section 2: Methods; Section 3: Results and Discussion; Section 4: Conclusions.

## 2. Methods

LCIA has become an international standardisation tool for environmental assessment [39,40]. Preliminary conditions need to be defined for every study: the functional unit and system boundary of the assessment were the six bridges and the SILAs of the corresponding communities. The assessment was conducted based on the LCIA, covering the whole of the life cycle. LCIA was analysed by using OpenLCA (Life cycle assessment) 1.10.1, LCCA by the budgetary estimate process, and SILA by OpenLCA1.10.3 (OpenLCA development team, Berlin, Germany) [14]. The three tools are relevant and systematic. The databases used in this study included Ecoinvent [41], Bedec [42], and Product Social Impact Life Cycle Assessment (PSILCA) [43]. See Section 2.1 and Section 2.2 for detailed modelling.

### 2.1. Modeling Analysis

The construction industry is the most active sector in both developed and developing countries, forming a high global consistency [44]. LCIA was included as a sustainable survey method, because it can systematically assess the environment in all directions and complete the selection of friendly products [45]. ISO has issued a series of 14,040 standards and International Life Cycle Data (ILCD) manuals to promote sustainable development [15,46].

#### 2.1.1. LCIA

The studied cases were six representative cable-stayed bridges, including South Tai Hu Lake Bridge (STHB), Shenzhen Bay Bridge (SZBB), New Bridge of Xishuangbanna Tropical Botanical Garden (BGNB), Cable-stayed Bridge of Changjiang West Road, Deyang City (CJWB), Hanjiang Highway Bridge, Xiantao City (XTHB), and Baishan Bridge, Baishan City (BSCB). Five of them adopted a reinforced concrete structure and one adopted a steel structure (the main beam of SZBB is constructed by welding and bolting steel components). All of them have a single tower. The length of the main bridge ranges from 136 to 410 m and all six bridges are Class I municipal highway bridges. Table 2 shows the detailed data.

According to ISO standards, and the requirement for the scope of strict assessment and examination of the life cycle of the bridge [47,48,49], the full life cycle of these six bridges was analysed in five stages: survey and design, material manufacturing, construction and installation, maintenance and operation, and disassembly and recycling. Since the cross section of the main girder of the bridge is variable, the calculation unit was based on 1 cubic meter. In order to achieve the rationality of the data comparison study and analysis, the study length of the six cable-stayed bridges was selected as a uniform 390 m to input relevant data (390 m including the main bridge and some auxiliary bridges).

Seven key impact categories, including energy use, ecotoxicity, acidification, eutrophication, climate change, particulate matter formation and ozone depletion, were determined through the comparative analysis of the oxidation separation of fossil materials and the European Union Product Environmental Footprint (EUPEF) [50,51,52]. Five of these seven categories were selected as the important goals of bridges’ LCIA: global warming potential (GWP), acidification potential (AP), free-water eutrophication potential (FEP), particulate matter formation potential (PMFP), including fumes and dust, and waste potential (WP).

The assessment and modelling method of LCIA has a midpoint and endpoint. Huijbregts et al. made a clear distinction and explanation in their reports ReCiPe 2008 and 2016 LCIA [53,54]. By comparing the advantages and disadvantages of the two modelling approaches [55], it was found that the midpoint modelling is more appropriate for stages, while the end-point modelling is more appropriate for intervals.

Major modelling formulas of LCIS:

Environmental impact contribution of transport vehicle:(1)$$\begin{array}{c} E_{m}=\sum_{i}^{j}\{K_{\mathrm{im}}\times \left[\sum_{i}^{j}\left(K_{i}+K_{2}+\cdots \cdots +K_{j}\right)\right]\times M\times (1+\alpha )\times V_{m}\times \lambda_{\mu }+\cdots \cdots \\ +K_{\mathrm{jm}}\times \left[\sum_{i}^{j}\left(K_{i}+K_{2}+\cdots \cdots +K_{j}\right)\right]\times M\times (1+\beta )\times V_{m}\times \lambda_{\mu }\} \end{array}$$
where $E_{m}$ = Environmental impact contribution of transport vehicle (kg); $K_{\mathrm{im},}K_{\mathrm{jm}}$ = Fuel consumption of vehicles i,j (L/100 km); $V_{m}$= Quantity of surveying vehicles i,j; α, β = Engine fuel loss of different types of vehicles (%); and $\lambda_{\mu }$ = Physical and chemical environmental emission coefficient of fuel μ (kg/kg) [56].

Environmental impact contribution of mechanical equipment:(2)$$M_{m}=\sum_{i}^{j}\{\left[G_{\mathrm{im}}\times \left(1+\alpha \right)\times T_{\mathrm{im}}\times \left(\lambda_{\mu }⊕\lambda_{\nu }\right)\right]+\cdots \cdots +\left[G_{\mathrm{jm}}\times \left(1+\alpha \right)\times T_{\mathrm{jm}}\times \left(\lambda_{\mu }⊕\lambda_{\nu }\right)\right]\}$$
where $M_{m}$ = Environmental impact contribution of mechanical equipment (kg); $G_{\mathrm{im}},G_{\mathrm{jm}}$ = Fuel consumption and power consumption of equipment i j (kg/h, kWh); $T_{\mathrm{im}}$ = Normal working hour of mechanical equipment (h); ⊕ = Logic “Or”; and $\lambda_{\nu }$ = Physical and chemical environmental emission coefficient of electric energy ν (kg/kg).

Environmental impact contribution of personnel:(3)$$P_{m}=W_{m}\times \lambda_{p}\times T_{p}$$
where $P_{m}$ = Environmental impact contribution of personnel (kg); $W_{m}$ = Total number of personnel (persons); $\lambda_{p}$ = Environmental impact coefficient of personnel (kg/working day/person); $T_{p}$ =Total working hours of personnel (working day).

Environmental impact contribution of office facilities:(4)$$W_{m}=\sum_{i}^{j}\left\{\left[F_{\mathrm{im}}\times T_{i}\times \left(1+L_{i}\right)\times \lambda_{i}]+\cdot \cdot \cdot \cdot \cdot \cdot [F_{\mathrm{jm}}\times T_{j}\times \left(1+L_{j}\right)\times \lambda_{j}\right]\right\}$$
where $W_{m}$ = Environmental impact contribution of office facilities (kg); $F_{\mathrm{im}},F_{\mathrm{jm}}$ = Power consumption of office facilities i, j (Kwh); $T_{i},T_{j}$ = Working hours of office facilities i ,j (h); and $L_{i},L_{j}$ = Electricity loss coefficient of facilities i, j (%).

SimaPro has been the world’s leading life cycle assessment (LCA) software package for 30 years; it is trusted by industry and academics in more than 80 countries [57]. OpenLCA can access the social and economic impact of 15 different life cycles. The software has been widely used in various industries and research fields in Europe, the United States, Japan and the rest of the world; it is supported by databases such as Ecoinvent, Bedec, Soca, bridge design, construction drawings, and published research results. 

#### 2.1.2. LCCA

LCCA of bridges mainly includes initial cost, cost of maintenance, repair and replacement, casualties of personnel or loss of goods during operation, road use cost, and indirect loss of socio-economic benefits [58,59]. In order to accurately estimate these costs, it is necessary to clarify the degradation rate of bridge components and build a correct model for the designated fatigue life index [60,61]. Table 2 shows the maintenance cycle. The core elements of LCCA are financial factors, inter-generational responsibility, environmental aspects and sustainability, realising the optimal balance between safety, economic efficiency, and sustainability [62].

LCCA was conducted in accordance with the process of highway engineering in China, as shown in Figure 1. It was of equal importance to determine the life cycle cost, cost benefit, or cost risk by considering a variety of ways of calculating cost benefit [58].
(5)$$E\left[C_{T}\left(\bar{x},\;T_{\mathrm{Ready}}\right)\right]=C_{i}\left(\bar{x}\right)+\sum_{t=1}^{T_{\mathrm{Ready}}}\left(\frac{\sum_{j=1}^{J}E\left[C_{\mathrm{Advisoryj}}\left(\bar{x},t\right)\right]+\sum_{k=1}^{K}E\left[C_{\mathrm{Assess}}\left(\bar{x},t\right)\right]+\sum_{l=1}^{L}E\left(C_{\mathrm{Mixed}}\left(\bar{x},t\right)\right)}{{\left(1+r\right)}^{t}}\right)$$
where $E\left[C_{T}\left(\bar{x},T_{\mathrm{Ready}}\right)\right]$ = LCCA cost in the preparatory stage (Chinese Yuan: CNY); $C_{i}\left(\bar{x}\right)$ = Direct cost in the preparatory stage (CNY);$\overset{J}{\underset{j=1}{\sum }}E\left[C_{\mathrm{Advisoryj}}\left(\bar{x},t\right)\right]$ = Consulting fee of the development organisation (CNY); $\overset{K}{\underset{k=1}{\sum }}E\left[C_{\mathrm{Assess}}\left(\bar{x},t\right)\right]$ = Impact assessment fee of the development organisation (CNY); $\overset{L}{\underset{l=1}{\sum }}E\left(C_{\mathrm{Mixed}}\left(\bar{x},t\right)\right)$ = Other costs incurred in the preparatory stage of the project, including expert review fee, transportation fee, approval procedure fee, office fee, labour fee for related personnel (CNY); and r = Discount rate (%).

The service rate for the project-bidding agency issued by National Development and Reform Commission is given by [63]:(6)$$\begin{array}{cc} C_{\mathrm{Bidding}\;\mathrm{Service}}= & \\ & \{\begin{array}{cc} 500\;\mathrm{million}\;\mathrm{CNY}\leq C_{\mathrm{Build}}\left(\bar{x},T_{\mathrm{End}}\right)\leq 1000\;\mathrm{million}\;\mathrm{CNY} & C_{\mathrm{BS}}=0.035\%*C_{\mathrm{Build}}\\ 1000\;\mathrm{million}\;\mathrm{CNY}<C_{\mathrm{Build}}\left(\bar{x},T_{\mathrm{End}}\right)\leq 5000\;\mathrm{million}\;\mathrm{CNY} & C_{\mathrm{BS}}=0.008\%*C_{\mathrm{Build}}\\ 5000\;\mathrm{million}\;\mathrm{CNY}<C_{\mathrm{Build}}\left(\bar{x},T_{\mathrm{End}}\right)\leq 10,000\;\mathrm{million}\;\mathrm{CNY} & C_{\mathrm{BS}}=0.006\%*C_{\mathrm{Build}}\\ 10,000\;\mathrm{million}\;\mathrm{CNY}<C_{\mathrm{Build}}\left(\bar{x},T_{\mathrm{End}}\right) & C_{\mathrm{BS}}=0.004\%*C_{\mathrm{Build}}\\ C_{\mathrm{Bidding}\;\mathrm{service}}=\mathrm{Maximum}\;\mathrm{amount}\;3.5\;\mathrm{million}\;\mathrm{CNY} & 3.0\;\mathrm{million}\;\mathrm{CNY},4.5\;\mathrm{million}\;\mathrm{CNY} \end{array} \end{array}$$

Costs of survey and design:(7)$$\begin{array}{cc} C_{\mathrm{Design}}\left(\bar{x},T_{\mathrm{End}}\right)= & \\ & \overset{T_{\mathrm{End}}}{\underset{t=\mathrm{Start}}{\sum }}\left\{\frac{\left[C_{\mathrm{Survey}}\left(\bar{x}\right)+\sum_{t_{1}}^{t_{\mathrm{End}}}\left(\bar{x},t_{\mathrm{Survey}}\right)\right]\left(1\pm \lambda_{h}\right)+\sum_{t_{1}}^{t_{\mathrm{End}}}\left(\bar{x},t_{\mathrm{Survey}}\right)+C_{\mathrm{Design}}\left(\bar{x}\right)+\sum_{t_{1}}^{t_{\mathrm{End}}}\left(\bar{x},t_{\mathrm{Design}}\right)}{{\left(1+r\right)}^{t}}\right\}\left[1\pm \left(C_{\mathrm{Float}}\right)\right]\left[1\pm \left(R_{t}\right)\right] \end{array}$$
where $C_{\mathrm{Design}}\left(\bar{x},T_{\mathrm{End}}\right)$ = LCCA cost in the stage of survey and design (CNY); $C_{\mathrm{Driect}}\left(\bar{x}\right)$, $C_{\mathrm{Design}}\left(\bar{x}\right)$= Direct cost in the stage of survey and design (CNY);$\overset{t_{\mathrm{End}}}{\underset{t_{1}}{\sum }}\left(\bar{x},t_{\mathrm{Survey}}\right),\overset{t_{\mathrm{End}}}{\underset{t_{1}}{\sum }}\left(\bar{x},t_{\mathrm{Design}}\right)$ =Indirect cost in the stage of survey and design (CNY); $R_{t}$= National tax rate (%); $C_{\mathrm{Float}}$= Adjustment range (%); and $\lambda_{h}$ = Adjustment coefficient [64].
(8)$$\lambda_{h}=\{\begin{array}{cc} T_{\mathrm{Local}}\geq 35\;{}^{\circ}C & \lambda_{h}=1.2\\ T_{\mathrm{Local}}\leq -10\;{}^{\circ}C & \lambda_{h}=1.2\\ 2000\;\mathrm{meters}\leq H_{\mathrm{Altitude}}\leq 3000\;\mathrm{meters} & \lambda_{h}=1.1\\ 3001\;\mathrm{meters}\leq H_{\mathrm{Altitude}}\leq 3500\;\mathrm{meters} & \lambda_{h}=1.2\\ 3501\;\mathrm{meters}\leq H_{\mathrm{Altitude}}\leq 4000\;\mathrm{meters} & \lambda_{h}=1.3\\ 4001\;\mathrm{meters}\leq H_{\mathrm{Altitude}} & \lambda_{h}\gg 1.3\left(\mathrm{Negotiated}\;\mathrm{price}\right) \end{array}$$
where $T_{\mathrm{Local}}$ = Ambient temperature of the place where the project locates (°C), and $H_{\mathrm{Altitude}}$ = Altitude of the place where the project locates (m).

Concerning the rate for the design and examination of construction drawings [63], it is charged by the budgetary investment ratio, thus the rate should not be higher than 2‰ of the budget amount of the project.

Construction costs:(9)$$\begin{array}{cc} C_{\mathrm{Build}}\left(\bar{x},T_{\mathrm{End}}\right)= & \\ & \overset{T_{\mathrm{Warranty}\;\mathrm{termination}}}{\underset{t=\mathrm{Start}}{\sum }}\left\{\frac{C_{\mathrm{Direct}\;\mathrm{cost}}+C_{\mathrm{Extra}\;\mathrm{charge}}+\left[\left(C_{\mathrm{Direct}\;\mathrm{cost}}+C_{\mathrm{Extra}\;\mathrm{charge}}\right)\;*\;C_{\mathrm{Profit}}\right]}{{\left(1+r\right)}^{t}}\right\}\left[1\pm \left(R_{t}\right)\right] \end{array}$$
where $C_{\mathrm{Build}}\left(\bar{x},T_{\mathrm{End}}\right)$ = LCCA cost in the stage of construction (CNY); $C_{\mathrm{Direct}\;\mathrm{cost}}$ = Direct cost of the project (CNY); $C_{\mathrm{Extra}\;\mathrm{charge}}$ = Indirect cost of the project (CNY); and $C_{\mathrm{Profit}}$ = Construction profits of the project (CNY).

Costs of maintenance and operation:

Global warming and extreme weather events have resulted in observable effects on people, the environment, and civil infrastructures [6]. Stewart et al. proposed four main factors for infrastructure corrosion and structural performance deterioration, including temperature [65]. Barbara Rossi et al. concluded that the total project cost decreases with the increase in the discount rate, and the period of investment return ranges between 18.5 and 24.2 years [66].

The six bridges are located in five economic belts. Climate, traffic density, traffic accidents, load effect of heavy-duty vehicles, and natural disasters (such as flooding, ice damage, freezing damage and mudslides) have different degrees of impact on the maintenance costs of bridges. The analysis was carried out according to the Chinese Code for Maintenance of Highway Bridges and Culverts (JTG H11-2004), as shown in Table 1 [67,68].

Costs of maintenance and repair: $C_{\mathrm{Maintenance}}\left(\bar{x},T_{100\mathrm{years}}\right)$=
(10)$$\{\begin{array}{cc} \overset{t=100\mathrm{years}}{\underset{t=1\mathrm{year}}{\sum }}\left(\frac{C_{\mathrm{Direct}\;\mathrm{cost}}+C_{\mathrm{Extra}\;\mathrm{charge}}}{{\left(1+r\right)}^{t}}\right)\left[1\pm \left(R_{t}\right)\right]\frac{T_{\mathrm{Total}\;\mathrm{maintanence}\;\mathrm{times}}}{T_{\mathrm{Times}\;\mathrm{of}\;\mathrm{cycle}}} & \mathrm{Maintenance}\;\mathrm{costs}\\ \overset{t=100\mathrm{years}}{\underset{t=1\mathrm{year}}{\sum }}\left(\frac{C_{\mathrm{Direct}\;\mathrm{cost}}+C_{\mathrm{Extra}\;\mathrm{charge}}}{{\left(1+r\right)}^{t}}\right)\left[1\pm \left(R_{t}\right)\right]\frac{T_{\mathrm{Strengthening}\;\mathrm{structure}\;\mathrm{times}}}{T_{\mathrm{Times}\;\mathrm{of}\;\mathrm{cycle}}} & \mathrm{Strengthening}\;\mathrm{structure}\;\mathrm{costs}\\ \overset{t=100\mathrm{years}}{\underset{t=1\mathrm{year}}{\sum }}\left(\frac{C_{\mathrm{Direct}\;\mathrm{cost}}+C_{\mathrm{Extra}\;\mathrm{charge}}}{{\left(1+r\right)}^{t}}\right)\left[1\pm \left(R_{t}\right)\right]\frac{T_{\mathrm{Emergency}\;\mathrm{repair}\;\mathrm{times}}}{T_{\mathrm{Times}\;\mathrm{of}\;\mathrm{cycle}}} & \mathrm{Emergency}\;\mathrm{repair}\;\mathrm{costs}\;\mathrm{of}\;\mathrm{road}\\ \overset{t=100\mathrm{years}}{\underset{t=1\mathrm{year}}{\sum }}\left(\frac{C_{\mathrm{Direct}\;\mathrm{cost}}+C_{\mathrm{Extra}\;\mathrm{charge}}}{{\left(1+r\right)}^{t}}\right)\left[1\pm \left(R_{t}\right)\right]\frac{T_{\mathrm{Routine}\;\mathrm{maintanence}\;\mathrm{times}}}{T_{\mathrm{Times}\;\mathrm{of}\;\mathrm{cycle}}} & \mathrm{Routine}\;\mathrm{maintenance}\;\mathrm{costs}\\ \overset{t=100\mathrm{years}}{\underset{t=1\mathrm{year}}{\sum }}\left(\frac{C_{\mathrm{Direct}\;\mathrm{cost}}+C_{\mathrm{Extra}\;\mathrm{charge}}}{{\left(1+r\right)}^{t}}\right)\left[1\pm \left(R_{t}\right)\right]\frac{T_{\mathrm{Intermediate}\;\mathrm{maintanence}\;\mathrm{times}}}{T_{\mathrm{Times}\;\mathrm{of}\;\mathrm{cycle}}} & \mathrm{Intermediate}\;\mathrm{maintenance}\;\mathrm{costs}\\ \overset{t=100\mathrm{years}}{\underset{t=1\mathrm{year}}{\sum }}\left(\frac{C_{\mathrm{Direct}\;\mathrm{cost}}+C_{\mathrm{Extra}\;\mathrm{charge}}}{{\left(1+r\right)}^{t}}\right)\left[1\pm \left(R_{t}\right)\right]\frac{T_{\mathrm{Heavy}\;\mathrm{maintanence}\;\mathrm{times}}}{T_{\mathrm{Times}\;\mathrm{of}\;\mathrm{cycle}}} & \mathrm{Heavy}\;\mathrm{maintenance}\;\mathrm{costs}\\ \overset{t=100\mathrm{years}}{\underset{t=1\mathrm{year}}{\sum }}\left(\frac{C_{\mathrm{Direct}\;\mathrm{cost}}+C_{\mathrm{Extra}\;\mathrm{charge}}}{{\left(1+r\right)}^{t}}\right)\left[1\pm \left(R_{t}\right)\right]\frac{T_{\mathrm{Mad}\;\mathrm{maintanence}\;\mathrm{times}}}{T_{\mathrm{Times}\;\mathrm{of}\;\mathrm{cycle}}} & \mathrm{Mad}\;\mathrm{improvement}\;\mathrm{costs} \end{array}$$
where $C_{\mathrm{Maintenance}}\left(\bar{x},T_{100\;\mathrm{years}}\right)$ = Costs of maintenance and operation (CNY); $T_{\mathrm{Number}\;\mathrm{of}\;\mathrm{cycle}}$ represents the days of each maintenance cycle (days); and $T_{\mathrm{Total}\;\mathrm{maintanence}\;\mathrm{times}},T_{\mathrm{Strengthening}\;\mathrm{structure}\;\mathrm{times}},T_{\mathrm{Emergency}\;\mathrm{repair}\;\mathrm{times}},\;T_{\mathrm{Routine}\;\mathrm{maintanence}\;\mathrm{times}},T_{\mathrm{Med}\;\mathrm{maintanence}\;\mathrm{times}},{\;T}_{\mathrm{Intermediate}\;\mathrm{maintanence}\;\mathrm{times}},{\;\mathrm{and}\;T}_{\mathrm{Heavy}\;\mathrm{maintanence}\;\mathrm{times}}$ represent the total time for maintenance (days), the total time for strengthening (days), the total time of emergency repair (days), the total time for routine maintenance (days), the total time for intermediate maintenance (days), the total time for heavy maintenance (days), and the total time for overhaul maintenance (days), respectively.

Costs of traffic accidents:

Civilian car ownership in China reached 232,312,300 units in 2018, increasing by 42.7% since 2015 [69]. Wang et al. analysed the severity of traffic accidents in China from a macro perspective, finding that the total fatality rate and man-made injury rate of highway traffic accidents from 2000 to 2016 increased by 19.0% and 63.7% [70]. Vlegel et al. found that the average per capita health care cost was EU 8200 and the productivity cost was EU 5900 [71]. Rukaibi et al. estimated that the average cost of a traffic accident in Kuwait was 9122 KD/crash (equivalent to EU 25,333.02) [72]. According to the data in the China Statistical Yearbook-2019, there were 244,937 traffic accidents in 2018, resulting in 63,194 deaths, 258,532 injuries, and direct property losses of CNY 1385 million [69].
(11)$$\begin{array}{cc} C_{\mathrm{Traffic}\;\mathrm{accident}}\left(\bar{x},T_{100\mathrm{years}}\right)\;= & \\ & \left\{ \begin{array}{c} \overset{t=100\mathrm{years}}{\underset{t=1\mathrm{year}}{\sum }}\left(\frac{C_{\mathrm{Human}\;\mathrm{costs}}+C_{\mathrm{Property}\;\mathrm{damage}}+C_{\mathrm{Other}\;\mathrm{related}\;\mathrm{losses}}}{{\left(1+r\right)}^{t}}\right)\left[{\left(1+e\right)}^{t}\right]\\ \overset{t=100\mathrm{years}}{\underset{t=1\mathrm{year}}{\sum }}C_{\mathrm{Human}\;\mathrm{costs}}=\overset{t=100\mathrm{years}}{\underset{t=1\mathrm{year}}{\sum }}\left(C_{\mathrm{Loss}\;\mathrm{of}\;\mathrm{productivity}}+C_{\mathrm{Quality}\;\mathrm{of}\;\mathrm{live}\;\mathrm{costs}}+C_{\mathrm{Medical}\;\mathrm{costs}}\right)\\ \overset{t=100\mathrm{years}}{\underset{t=1\mathrm{year}}{\sum }}C_{C_{\mathrm{Property}\;\mathrm{damage}}}=\overset{t=100\mathrm{years}}{\underset{t=1\mathrm{year}}{\sum }}\left(C_{\mathrm{Vehicle}\;\mathrm{damage}\;\mathrm{costs}}+C_{\mathrm{Non}\;\mathrm{vehicle}\;\mathrm{damage}\;\mathrm{costs}}\right)\\ \overset{t=100\mathrm{years}}{\underset{t=1\mathrm{year}}{\sum }}C_{\mathrm{Other}\;\mathrm{related}\;\mathrm{losses}}=\overset{t=100\mathrm{years}}{\underset{t=1\mathrm{year}}{\sum }}\left(C_{\mathrm{Adminstration}\;\mathrm{costs}}+C_{\mathrm{Enviromental}\;\mathrm{costs}}+C_{\mathrm{Travel}\;\mathrm{delay}\;\mathrm{costs}}\right) \end{array}\right. \end{array}$$
where $C_{\mathrm{Traffic}\;\mathrm{accident}}\left(\bar{x},T_{100\mathrm{years}}\right)$ = Cost of traffic accidents (CNY); $C_{\mathrm{Human}\;\cos\mathrm{ts}}$;${\;C}_{\mathrm{Property}\;\mathrm{damage}};$
$C_{\mathrm{Other}\;\mathrm{related}\;\mathrm{losses}}$ = Human costs (CNY); property damage (CNY); other related losses (CNY); and e = Economic growth rate (%).

The six bridges studied are municipal highway bridges and no traffic tolls were charged during the operation.

The total costs required in the stage of maintenance and operation are the sum of Equations (10) and (11).

Disassembly costs:

The cable-stayed bridges will be disassembled at the expiration of their designed service life. The modelling of incurred costs was subject to Eq. (4). The materials to be demolished include broken concrete, scrap steel and waste. Construction wastes dumped and stacked in the natural environment without authorisation are one of the sources of environmental pollution [73]. In recent years, countries all over the world have been using recycled materials for sustainable development and steel is re-smelted for recycling [74,75].

Recycling cost of waste and scraps:(12)$$C_{\mathrm{Recycling}}\left(\bar{x},T_{\mathrm{Recycling}}\right)\;=\overset{t=\mathrm{New}\;\mathrm{product}}{\underset{t=\mathrm{Sec}\mathrm{ondary}\;\mathrm{processing}}{\sum }}\left[\frac{C_{\mathrm{Waste}\;\mathrm{concrete}}{\;*\;u}_{\mathrm{Concrete}}{\;*\;C}_{\mathrm{Post}-\mathrm{processing}\;\mathrm{costs}+}C_{\mathrm{Waste}\;\mathrm{steel}}{\;*\;u}_{\mathrm{Steel}}{\;*\;C}_{\mathrm{Steelmaking}\;\mathrm{costs}}}{{\left(1+r\right)}^{t}}\right]$$
where $C_{\mathrm{Recycling}}\left(\bar{x},T_{\mathrm{Recycling}}\right)$ = Recycling costs of waste and scraps (CNY); $C_{\mathrm{Waste}\;\mathrm{concrete}}$ = Quantity of demolished concrete waste (kg);${\;u}_{\mathrm{Concrete},u_{\mathrm{steel}}}$ = Recovery rate of concrete and steel waste (%); and $C_{\mathrm{Steelmaking}\;\mathrm{costs}},C_{\mathrm{Post}-\mathrm{processing}\;\mathrm{cost}}$ ”51” = Cost of recycling and disposal (CNY/kg).

#### 2.1.3. SILA

SILA witnessed its heyday from 1970 to 1980 and has been widely practiced in many fields around the world [76]. Social impact assessment comprises analysing, monitoring, and managing the social impacts of a project to bring about a more sustainable and equitable biophysical and human environment [77]. However, the assessment criteria and the quality of collected data are affected due to the limited resources of social assessment and the limited ability of regulatory agencies to control the management system [78,79].

PSILCA and USDA data and the Social Hotspots Database (SHDB) were used in this study to assess the research on sustainable social pillars [80,81]. The PSILCA database features the latest data sources, the original data sources and the quality assessment of risk data. Furthermore, the social contact messages from the PSILCA database can be associated with each other in the manner of SOCA (SOCA is an add-on for the Ecoinvent database, containing social inventory data based on PSILCA.) via Green Delta. The processes that are identical to those in environmental assessment can be used for social assessment, thus realising the coherence of the entire assessment (show in Figure 2). SILA uses input data from the LCIA for environmental and social assessment and determines 54 quantitative and qualitative indexes for 18 categories [82]. Five of the analysis indexes are closely related to the community stakeholders, according to the location where the six bridges are located and can be used as the factors for the social impact analysis. The five indexes are fatal accidents (FA), international migrant workers (IMW), youth illiteracy (YI), corruption (C), and sanitation coverage (SC).

According to the location of the six bridges in the region, the five indexes selected are closely linked to community stakeholders and can be used as factors for social impact analysis.

### 2.2. Research Process

The six cable-stayed bridges across five geographical zones of China (Northeast China, East China, Central China, South China, and Southwest China) and six provinces (Zhejiang, Guangdong, Sichuan, Hubei, Yunnan and Jilin) were selected as the objects of study [83]. They are important in terms of geographical location, economic value, environmental impact, and social assessment, becoming the strong backing for this study, as shown in Figure 3.

#### 2.2.1. LCIA

General information about the six bridges is shown in Table 2. All of these bridges have been completed and put into operation. They are the main highway bridges of the cities where they are located.

The Chinese government classifies cities by criteria including the agglomeration degree of commercial resources, urban pivotability, resident activeness, lifestyle diversity and future plasticity [84]. Among these six cable-stayed bridges, STHB is located in a third-tier city, SZBB in a first-tier city, BGNB in a fourth-tier city, CJWB in a fourth-tier city, XTHB in a fifth-tier city, and BSCB in a fifth-tier city.

They were designed by six design institutes in different regions, which are between 84 and 2380 km away from the project sites. The surveying equipment used was self-owned, calibrated equipment with high precision, which needed to be transported by truck to the project site. The expressway is the preferred mode of transport, but rail travel should be adopted if the transport distance is larger than 500 km. The development organisation was not allowed to use self-produced concrete for cable-stayed bridges, because the bridges are municipal works. All concrete used for the cable-stayed bridges had to be purchased as commercial concrete. Concrete is classified into C55, C50, C40, C30, C25 and C20. SZBB is a steel bridge, using 374 m^3^ of precast blocks of commercial concrete for the bridge deck.

During the construction, the materials were mainly transported and hoisted by a tower crane, a 25 T/50 T truck crane, and a floating crane (for the sections across the river), because the main tower of the cable-stayed bridge was too high. The main beam of SZBB is made of Q345-C low alloy steel and the accessory structures are made of Q235-B steel. The components and parts were connected by high-strength bolts and welding. The bridge was divided into 31 beam sections, which were manufactured in the factory and then transported to the bridge position by barges. The floating crane and land cranes worked together to lift and install these sections in the right place. The other five cable-stayed bridges adopted reinforced concrete structures. The main towers were subject to cast-in-place construction with creeping formwork by sections. The main beams were subject to cast-in-place construction with a sliding formwork using the full framing method. The details are shown in Table 3.

BSCB is located in Baishan City, Jilin Province. The construction environment is affected by the local climate. The local temperature in winter can be as low as −42 °C, with an annual average temperature of 4.6 °C [85]. Construction has to be stopped in October every year and can restart again by the end of April of the next year. The affected construction duration reaches 210 days a year.

The operation stage is the key period for the environmental impact contribution of bridges. A large number of vehicles will emit exhaust gases within the 100 years of service life, causing severe environmental pollution. Exhaust gas pollution is the key to research on LCIA. Dargay et al. concluded that the automobile saturation in China is 807 vehicles for every 1000 persons [86], which is set as the upper limit of the number of vehicles in each region. According to the study by Wu et al., car ownership will grow up to 4.8% in 2030, with the growth rate in 2050 being 2.9%, reaching 455 vehicles for every 1000 persons [87]. The traffic volume in 100 years is determined according to the comprehensive data analysis of the China Statistical Yearbook [88], as shown in Figure 4.

Establish a traffic flow analysis model: (13)$$\begin{array}{cc} N_{V}(x,T_{\mathrm{SO}}^{\mathrm{SD}})\;= & \\ & \left\{ \begin{array}{ccc} T_{C}^{D} & T_{C}=\mathrm{Completion}\;\mathrm{report}\;\mathrm{query},T_{D}=\mathrm{Design}\;\mathrm{time}\left(100\;Y\right) & ①\\ N_{P_{T_{C}}}^{P_{T_{D}}}\left(\lambda 1_{\mathrm{GR}}\right) & P_{C}=\left[88\right]\;I,P_{D}=\left[96\right]\;I\left(P_{2050\;Y}^{1.42\;B},P_{2060\;Y}^{1.365\;B},P_{2110\;Y}^{1.0\;B}\right) & ②\\ N_{H}\left[T_{\mathrm{Quantity}\;\mathrm{of}\;\mathrm{the}\;Y}^{\mathrm{Urban}\;H\;\mathrm{in}\;100\;Y}\left(\lambda 2_{\mathrm{GR}}\right)\right] & T_{\mathrm{Quantity}\;\mathrm{of}\;\mathrm{the}\;Y}=\left[96\right]I,T_{\mathrm{Urban}\;H\;\mathrm{in}\;100\;Y}=\mathrm{CA}\;\mathrm{based}\;\mathrm{on}\;\lambda & ③\\ N_{N\;\mathrm{of}\;V}\left[v_{C}^{100\;Y}\left(\lambda 3_{\mathrm{GR}}\right)\right] & V_{N}=\left[86\right]\;I\left(\mathrm{GA}=\frac{455\;V}{1000\;\mathrm{persons}}\left[87\right]\right),\lambda 3_{\mathrm{GR}}\left(\lambda 3_{2030\;Y}^{4.8\%},\lambda 3_{2050\;Y}^{2.9\%},\right) & ④\\ N_{1000\;\mathrm{persons}}^{N\;\mathrm{of}\;V} & N=\mathrm{The}\;\mathrm{above}\;①②③④\;\mathrm{CA} & ⑤\\ N_{\mathrm{Every}\;Y}^{\mathrm{The}\;N\;\mathrm{of}\;V\;\mathrm{passing}\;\mathrm{on}\;\mathrm{the}\;\mathrm{bridge}}=④\times L_{\mathrm{Bridge}} & N=\mathrm{The}\;\mathrm{above}\;⑤\;\mathrm{CA} & ⑥ \end{array}\right. \end{array}$$
where B = Billion; CA = Calculated; C = Completed; D = Disassembly; GA = Greatest amount; GR = Growth rate; H = Highways; I =Inquiry; N = Number; P = Population; SD = Start disassembly; SO = Start operation; V = Vehicles; and Y = Years. (Note: this abbreviation is only used in Equation (13)).

Figure 4 and Equation (13) show that the traffic volume of SZBB and BGNB is 2 to 5 times that of the traffic volume of the other four bridges, which will affect the subsequent environmental pollution data of the bridges. After 2000, infrastructure expenditure in China accounted for approximately 6.5% of gross domestic product (GDP), much higher than the average level of 4% in other developing countries. After 2009, coastal provinces and cities increased investment in infrastructure (including energy, transportation, telecommunications, water and sewage treatment), reaching 15–20% of GDP [89].

After the expiration of the operation stage, the cable-stayed bridges enter the disassembly stage. These bridges will be demolished by mechanical disruption because blasting demolition has many safety-impacting factors and these bridges are all located in urban areas. The scrapped steel materials will be transported to steel works for recycling. Concrete blocks will be transported to the production plants of reclaimed materials for crushing and classification. All of the remaining waste will be transported to the waste treatment plant for recycling.

#### 2.2.2. LCCA

All of these cable-stayed bridges are municipal works, so the construction costs are analysed based on Engineering Standards for China’s Transportation Industry, JTG 3830-2018 Measures for Preliminary Estimate/Budgeting of Highway Projects, and JTG/T 3831-2018 Norms for Preliminary Estimate of Highway Projects [90].

The construction cost is first calculated by Equation (9), in accordance with design drawings, bill of quantities, and norms for preliminary estimates of highway projects. As shown in Table 4, the construction costs of the cable-stayed bridges were: CNY 72,055,116.25 for STHB, CNY 103,996,538.70 for SZBB, CNY 18,803,871.58 for BGNB, CNY 24,721,480.22 for CJWB, CNY 47,164,942.89 for XTHB, and CNY 37,812,245.23 for BSCB, respectively.

In the operation stage, aging parts and components need to be repaired and replaced in the bridges. Table 1 presents the maintenance and repair cycles of the main components. The costs generated by multiple replacements will be included in the costs for the construction stage, and the economic growth coefficient can then be considered.

The costs of traffic accidents are mainly used to analyse losses caused by traffic accidents and related expenses. According to the Chinese transportation statistics [32], the incidence of traffic accidents from 2001 to 2018 dropped by 25.7%, resulting in the reduction in property losses by 29.3%. After 2014, the annual reduction rate of traffic accidents stayed between 0.4% and −0.7%, and the property losses remained at CNY 5600 per accident.

As shown in Table 5, LCCA was conducted in three stages. The first stage covered the years from 2003 to 2018. The costs of traffic accidents were analysed based on the existing data. The coefficient for the growth or reduction rate of traffic accidents in 15 years, and the annual average number of traffic accidents were also determined. The second stage covered the years from 2019 to 2030. In 2030, the population of China will reach its peak and so will the level of car ownership (Figure 4). The population and car ownership will begin to decline after 2031 and the accident rate will tend to be stable.

#### 2.2.3. SILA

As shown in Figure 2, SILA was also conducted in five stages. The impact of the bridges on communities was analysed for all aspects, from the design stage to the final disassembly stage. The International Finance Corporation’s Performance Standards on Social and Environmental Sustainability (IFC 2012a) was taken as the reference. These Standards has become globally recognised good practice for handling environmental and social risk management and has been adopted by more than 80 leading banks as the “gold standard” for guiding project development [92,93]. The Standards formulate eight performance standards, including social and environmental assessment and management systems, labour and working conditions, pollution prevention and abatement, community health, safety and security, land acquisition and involuntary resettlement, biodiversity conservation and sustainable natural resource management, indigenous peoples, and cultural heritage. Based on the characteristics of Chinese communities (aboriginals will not be considered, because there are no aboriginals in the communities where cable-stayed bridges are located, and cultural heritage will also not be considered, because there is no newly-built cultural heritage in the construction areas), and the latest assessment factors in the PSILCA database, five assessment standards were selected as the research parameters, in accordance with the conclusions of comprehensive analysis (see Figure 2).

## 3. Results and Discussion

### 3.1. LCIA

According to our findings (shown in Table 6), the GWPs of six bridges are the main sources of environmental pollution, accounting for over 92% of the total pollution of each bridge. This is why the authors chose these five parameters in the long-term research. Effective control of GWP is the top priority for alleviating global pollution.

Figure 5 shows the environmental impact contributions of the six cable-stayed bridges, in the maintenance and operation stage, as follows: STHB = 209,488.94 tonnes > XTHB = 133,511.65 tonnes > BSCB = 126,010.36 tonnes > CJWB = 648,518 tonnes > BGNB = 49,735.66 tonnes > SZBB = 1230.24 tonnes.

An interesting research finding is that the main beam of SZBB is a steel structure,$\;Environmental\;impact\;contribution_{material\;manufacturing\;stage}$ > $Maintenance\;and\;operation\;stage_{material\;manufacturing\;stage}$, which is 40,327.22 tonnes and accounts for 49.73% of the total contribution of SZBB. This finding also proves that the environmental impact contribution of the steel bridge mainly comes from the material manufacturing stage and the construction and installation stage, accounting for 83.82% of the total contribution. Although there is a huge difference between the environmental impact contribution of a steel bridge and that of a concrete bridge, the total environmental impact contribution of the two kinds of bridges are approximate to each other.

### 3.2. Comparison

The differences in the durability of building materials and standards between Europe and China result in a difference in the life span of bridges, and the difference is mainly manifested in the service life of concrete; the warranty period of concrete for stay cables in Europe is 100 years, while in China, it is 20 or 50 years [67,94].

Thus, a large amount of maintenance and replacement work is required, resulting in great changes in environmental pollution values during the maintenance period.

Table 7 shows the environmental impact contribution values of five impact factors in the maintenance stage. Subject to the European and Chinese design standards, the maximum value falls on $GWP_{European\;standard}$ = 5343.68 tonnes for SZBB and $GWP_{Chinese\;standard}$ = 19,736.99 tonnes for STHB. Interestingly, the value of SZBB’s steel structure under the European standard is 10,824.72 tonnes greater than that under the Chinese standard. The difference in the design life of the materials leads to 33- to 73-fold differences, in terms of the environmental pollution value in the maintenance stage, and this is just a comparison analysis for one stage.

Figure 6 shows the difference in the environmental pollution value for the six bridges under five environmental impact factors and subject to two standards. The replacement times of the exposed stable cables and concrete of the cable-stayed bridges in the 100 years of the service life increases with time, resulting in an increase in GWP by 3249~15761 tonnes, particularly the GWP of the steel bridge at SZBB, which reduces by 4568 tonnes. The pollution contributions of the six cable-stayed bridges increase by 549,412.2 tonnes in total, which is an amazing figure.

### 3.3. LCCA

The conclusions of LCCA are shown in Table 8. The bridges selected in the case analyses are located in China, so the norms for Chinese highways were used in each analysis. For the cable-stayed bridges with reinforced concrete structures, the cost ratio of the maintenance and operation stage remains between 49% and 64%. However, the cost of steel bridges in the construction stage accounts for 63.2% of the total expenses because of the high investment cost. The maintenance cost of the steel bridge is 30% lower than that of the concrete bridge. The main reason is that the steel structure is superior to the concrete structure in terms of durability.

As shown in Figure 7, the maintenance cost of STHB is CNY 120 million, which is 1.8 times the construction cost. The maintenance costs of BGNB, CJWB, XTHB and BSCB are 2.0 to 2.3 times their construction costs. For the cable-stayed bridges with the reinforced concrete structure, the stay cables and concrete need to be replaced two to five times, because their service life and durability ranges between 20 and 50 years. Costs for multiple replacement events are the primary reason for the excessive maintenance costs, so the key to reducing costs is to improve the service life of materials.

### 3.4. SILA

SILA was conducted for the six cable-stayed bridges from four categories, including the population impact, community system, social resources and economic development. Five impact factors were selected according to the classification.

Table 9 shows some of the SILA values for the six bridges. For each cable-stayed bridge, corruption > sanitation coverage > fatal accidents > international migrant workers > youth illiteracy.

As shown in Figure 8, the values of five impact factors in each stage of the six cable-stayed bridges are ranked as follows:$$Numbers_{Construction\;and\;installation\;stage}>Numbers_{Decommissioning\;and\;dismantling\;stage}>\;Numbers_{Structural\;materials\;processing\;and\;construction\;stage}>Numbers_{Design\;stage}>Numbers_{Maintenance\;and\;operation\;stage}.$$

### 3.5. Deepen the Analysis

#### 3.5.1. Economic Evaluation

As shown in Figure 9, the bridges with the peak value of GDP in 10 years are SZBB and STHB (Government, n.d.); the bridges with the peak value of LCIA are STHB and XTHB; the bridges with the peak values of LCCA and SLCA are STHB and SZBB. The analysis concludes that the environmental pollution, production cost and social impact generated by infrastructure in developed regions increase accordingly. In particular, there is a complementary relationship between GDP and the emissions of environmentally polluting gases. The constant emission load of environmental pollution gases under GDP growth signifies that the current energy technologies must be replaced with renewable energy resources, and/or more energy-efficient production technologies must be adopted [95].

#### 3.5.2. Modelling and Discussion

Definition of Markov chain: assuming that $X_{1},X_{2},\cdots \cdots X_{n}$ is the discrete sequence of random influence variables, abbreviated as $\left\{X_{n}\right\}$, the state space of the entire $\left\{X_{n}\right\}$ is denoted as $E=\left\{x_{1},x_{2},\cdots \cdots x_{n}\right\}$; if any impact factor is subject to $x_{i1}x_{i2},\cdots \cdots x_{\mathrm{in}}E$, then $P\left(X_{n+1}\right)=\left(x_{i_{n+1}}|X_{1}=x_{i_{1}},\cdots \cdots X_{n}=x_{i_{n}}\right)$.

The impact matrix is established based on the definition,
(14)$$K_{h}=\{\begin{array}{cccc} x_{11}\left(h_{1}\right) & x_{12}\left(h_{1}\right)\cdots \cdots & x_{1m}\left(h_{1}\right) & h_{1}={\mathrm{GDP}}_{\mathrm{variables}}\\ x_{21}\left(h_{2}\right) & x_{22}\left(h_{2}\right)\cdots \cdots & x_{2m}\left(h_{2}\right) & h_{2}={\mathrm{GWP}}_{\mathrm{variables}}\\ \vdots & \vdots & \vdots & \vdots \\ \vdots & \vdots & \vdots & \vdots \\ x_{n1}\left(h_{k}\right) & x_{n2}\left(h_{k}\right)\cdots \cdots & x_{\mathrm{nm}}\left(h_{k}\right) & h_{k}=H_{\mathrm{variables}} \end{array}$$
where $K_{h}$ = conclusion of the infrastructure’s comprehensive impact assessment.

According to Equation (13),
$$K_{\mathrm{Six}\;\mathrm{bridges}}=\left[\begin{array}{cccccccc} \mathrm{GDP} & \mathrm{LCIA} & \mathrm{LCCA} & \mathrm{SLCA}-1 & \mathrm{SLCA}-2 & \mathrm{SLCA}-3 & \mathrm{SLCA}-4 & \mathrm{SLCA}-5\\ 4766 & 2943 & 2143 & 559 & 318 & 284 & 1189 & 885\\ 8169 & 811 & 1645 & 473 & 269 & 243 & 1007 & 750\\ 1510 & 718 & 665 & 65 & 37 & 33 & 139 & 103\\ 3384 & 832 & 1014 & 92 & 52 & 47 & 196 & 146\\ 3289 & 1726 & 1551 & 283 & 162 & 146 & 604 & 450\\ 1453 & 1555 & 1261 & 141 & 80 & 72 & 300 & 223 \end{array}\right]\;K_{\mathrm{Six}\;\mathrm{bridges}1}=\left[\begin{array}{cccccc} \mathrm{GDP} & \mathrm{LCIA} & \mathrm{LCCA} & \mathrm{SLCA}-1 & \mathrm{SLCA}-2 & \mathrm{SLCA}-3\\ 4766 & 2943 & 2143 & 559 & 318 & 284\\ 8169 & 811 & 1645 & 473 & 269 & 243\\ 1510 & 718 & 665 & 65 & 37 & 33\\ 3384 & 832 & 1014 & 92 & 52 & 47\\ 3289 & 1726 & 1551 & 283 & 162 & 146\\ 1453 & 1555 & 1261 & 141 & 80 & 72 \end{array}\right]\;K_{\mathrm{Six}\;\mathrm{bridges}1}=\left[\begin{array}{cccccc} 4766 & 2943 & 2143 & 559 & 318 & 284\\ 8169 & 811 & 1645 & 473 & 269 & 243\\ 1510 & 718 & 665 & 65 & 37 & 33\\ 3384 & 832 & 1014 & 92 & 52 & 47\\ 3289 & 1726 & 1551 & 283 & 162 & 146\\ 1453 & 1555 & 1261 & 141 & 80 & 72 \end{array}\right]\;\mathrm{Assuming}\;\left|K_{\mathrm{Six}\;\mathrm{bridges}1}-\lambda_{1}E\right|=\left|\begin{array}{cccccc} 4766-\lambda_{1} & 2943 & 2143 & 559 & 318 & 284\\ 8169 & 811-\lambda_{1} & 1645 & 473 & 269 & 243\\ 1510 & 718 & 665-\lambda_{1} & 65 & 37 & 33\\ 3384 & 832 & 1014 & 92-\lambda_{1} & 52 & 47\\ 3289 & 1726 & 1551 & 283 & 162-\lambda_{1} & 146\\ 1453 & 1555 & 1261 & 141 & 80 & 72-\lambda_{1} \end{array}\right|=\sum_{1}^{6}K_{\mathrm{bridges}1}$$

If the diagonal method is used, then (14) = (4766 − λ_1_) × (811 − λ_1_) × (655 − λ_1_) × (92 − λ_1_) × (162 − λ_1_) × (72 − λ_1_) − 433287870784λ_1_ − 5454599392867510 = 0, $\lambda_{1}$=$\overset{7}{\underset{1}{\sum }}\left(12588+4766+811+665+92+162+72\right)/7\;$= 2736.7 ≈ 2737.
$$K_{\mathrm{Six}\;\mathrm{bridges}2}=\left[\begin{array}{cccccc} 4766 & 2943 & 2143 & 559 & 1189 & 885\\ 8169 & 811 & 1645 & 473 & 1007 & 750\\ 1510 & 718 & 665 & 65 & 139 & 103\\ 3384 & 832 & 1014 & 92 & 196 & 146\\ 3289 & 1726 & 1551 & 283 & 604 & 450\\ 1453 & 1555 & 1261 & 141 & 300 & 223 \end{array}\right]$$
(15)$$\mathrm{Assuming}\;\left|K_{\mathrm{Six}\;\mathrm{bridges}2}-\lambda_{2}E\right|=\left[\begin{array}{cccccc} 4766-\lambda_{2} & 2943 & 2143 & 559 & 1189 & 885\\ 8169 & 811-\lambda_{2} & 1645 & 473 & 1007 & 750\\ 1510 & 718 & 665-\lambda_{2} & 65 & 139 & 103\\ 3384 & 832 & 1014 & 92-\lambda_{2} & 196 & 146\\ 3289 & 1726 & 1551 & 283 & 604-\lambda_{2} & 450\\ 1453 & 1555 & 1261 & 141 & 300 & 223\lambda_{2} \end{array}\right]=\;\sum_{1}^{6}K_{\mathrm{bridges}2}$$

If the diagonal method is used, then (15) (4766 − $\lambda_{2}$) × (811 − $\lambda_{2}$) × (665 − $\lambda_{2}$) × (92 − $\lambda_{2}$) × (604 − $\lambda_{2}$) × 223$\lambda_{2}$ − 147825193568210000$\lambda_{2}$ − 1046549405522410 = 0, $\lambda_{2}$ = $\overset{7}{\underset{1}{\sum }}\left(82565+4766+811+665+92+604+223\right)/7$=12818.

Based on Equations (14) and (15), we can conclude that the most reasonable impact range is 2737 < $K_{\mathrm{Six}\;\mathrm{bridges}}$ < 12818.

According to Figure 10, five-point positions are located in the reasonable comprehensive evaluation range. The five points are Point ② and ⑤ of STHB, Point ① of SZBB, Point ③ of CJWB, and Point ④ of XTHB.

## 4. Conclusions

The manuscript proposes a comprehensive and effective sustainability assessment method and establishes an assessment framework and modelling theory for complex structural bridges (cable-stayed bridges) in terms of environment, economy, and social impact. Through the comprehensive evaluation of six highway cable-stayed bridges in five provinces of China in the whole life cycle (from cradle to grave), the conclusion is drawn. GWP is the main source of environmental pollution in LCIA, accounting for more than 92% of the emissions of each bridge, which are concentrated in the maintenance stage. In LCCA, the proportion of maintenance stage cost is 49–64%. In SILA, the corruption value has the greatest influence, accounting for 36% of the total amount. The SZBB steel structure bridge is special: GWP accounts for 50% in the LCIA material stage and 63.2% in the LCCA construction stage.

In view of the high pollution and high cost in the maintenance stage, the conclusion shows that it is closely related to the design standard and service life of the materials. It is found that the difference in LCIA between Europe and China is 33~73-fold, which is due to the difference in the replacement period between the main girder and stay cable of 80 years and 50 years/cycle. More interestingly, the LCIA value of SZBB in Europe is higher than that in China by 10,824.7 tonnes, because the maintenance period of steel structure differs by 15 years/cycle. The differences in the above conclusions are closely related to regional population density, vehicle ownership and traffic frequency, which is one of the research directions in the future.

Finally, to obtain the relationship between GDP and sustainable impact, the comprehensive evaluation coefficient of the influence matrix is established by using discrete mathematics for multi factor decision-making, and the reasonable range of 2737~12,818 (The theoretical judgment standard of innovation) between China’s five major economic regional bridges and regional GDP is analysed.

This study aims to propose a complete method for assessing the sustainability of bridges. This article provides important knowledge for preliminary decisions in the construction of bridges as well as how to mitigate the loads of the three pillars. The limitation of the study is that there is no questionnaire survey in the social impact assessment, and it is impossible to compare and analyse whether there is a big difference between the conclusion and the actual impact. Future research directions need to strengthen the resilience analysis of evaluating the impact of the construction industry on society, and the mutual promotion and optimization of the GDP influencing factors and sustainable development.

## Figures and Tables

**Figure 1 ijerph-18-00122-f001:**
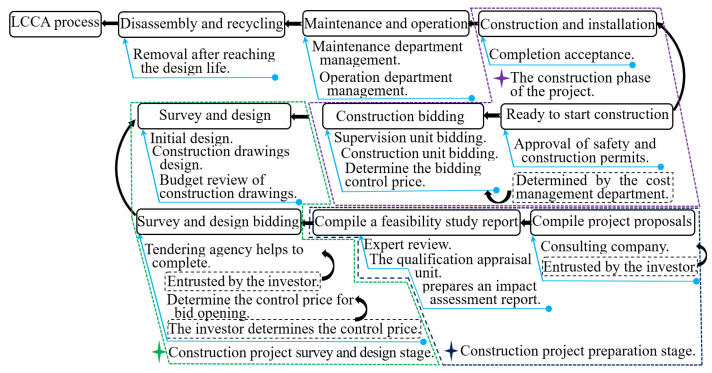
Basic procedure flow chart of highway engineering construction.

**Figure 2 ijerph-18-00122-f002:**
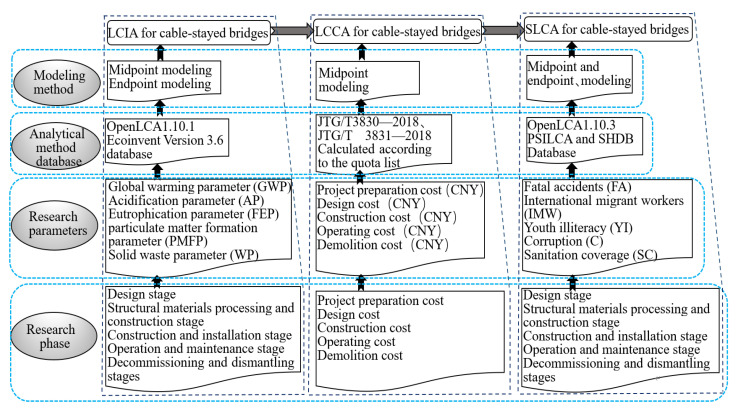
Schematic diagram of the LCIA, LCCA, and SILA analysis process.

**Figure 3 ijerph-18-00122-f003:**
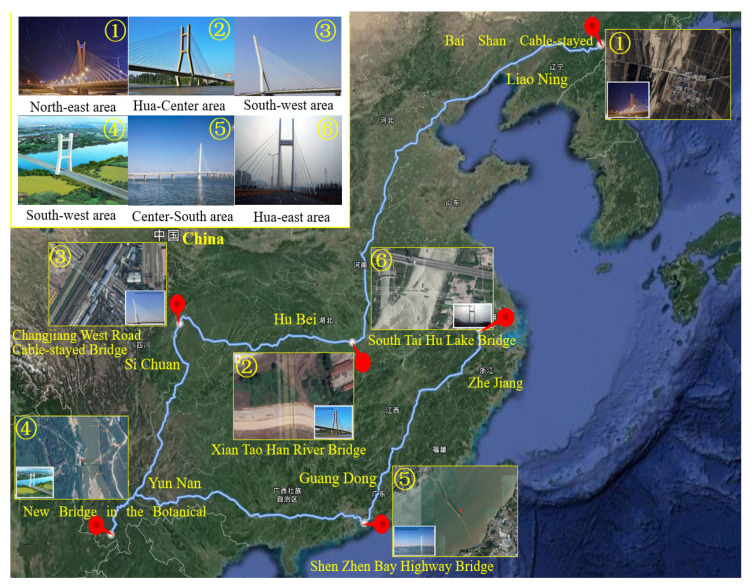
Schematic diagram of cable-stayed bridge regional distribution [83].

**Figure 4 ijerph-18-00122-f004:**
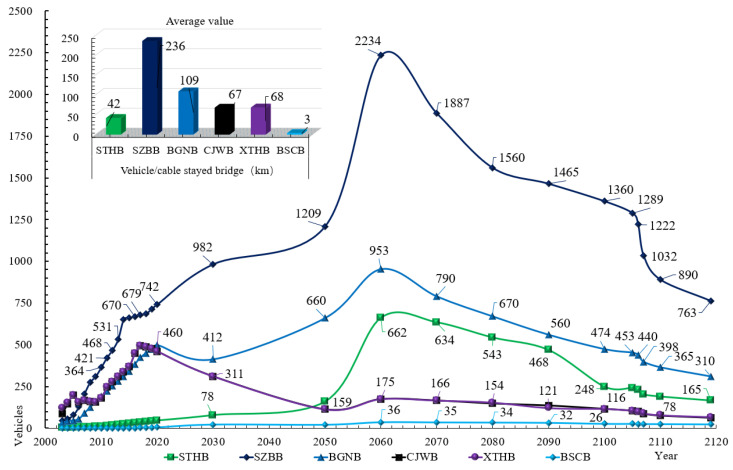
Schematic diagram of the number of vehicles driving on six cable-stayed bridges.

**Figure 5 ijerph-18-00122-f005:**
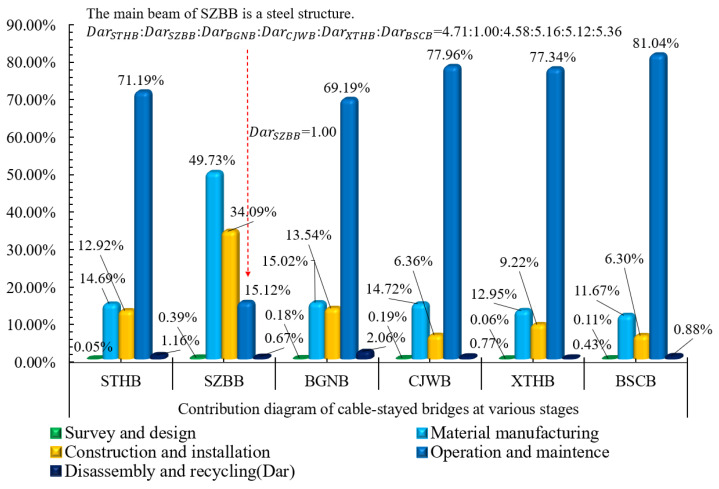
Environmental impact contribution diagrams of six cable-stayed bridges at various stages.

**Figure 6 ijerph-18-00122-f006:**
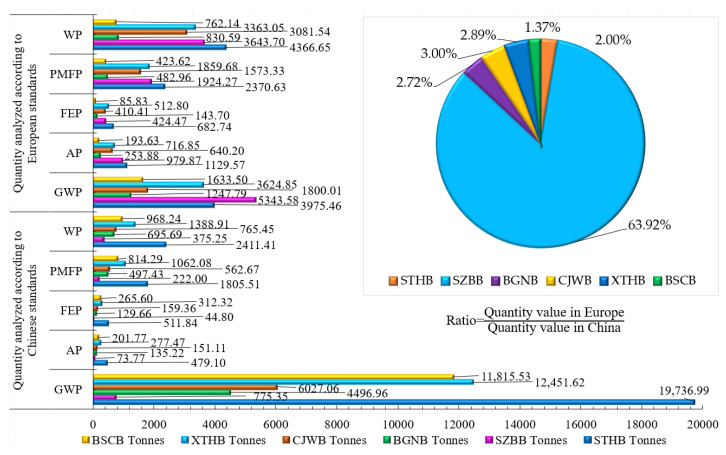
Environmental impact contribution diagrams of six cable-stayed bridges at various stages.

**Figure 7 ijerph-18-00122-f007:**
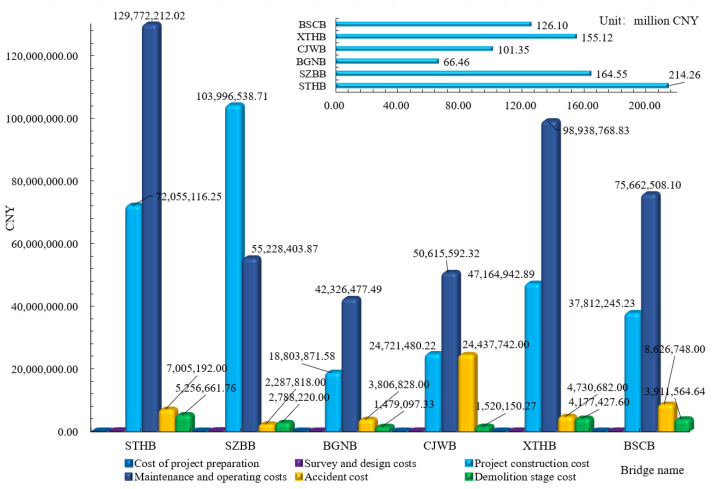
The cost diagram of six cable-stayed bridges at different stages.

**Figure 8 ijerph-18-00122-f008:**
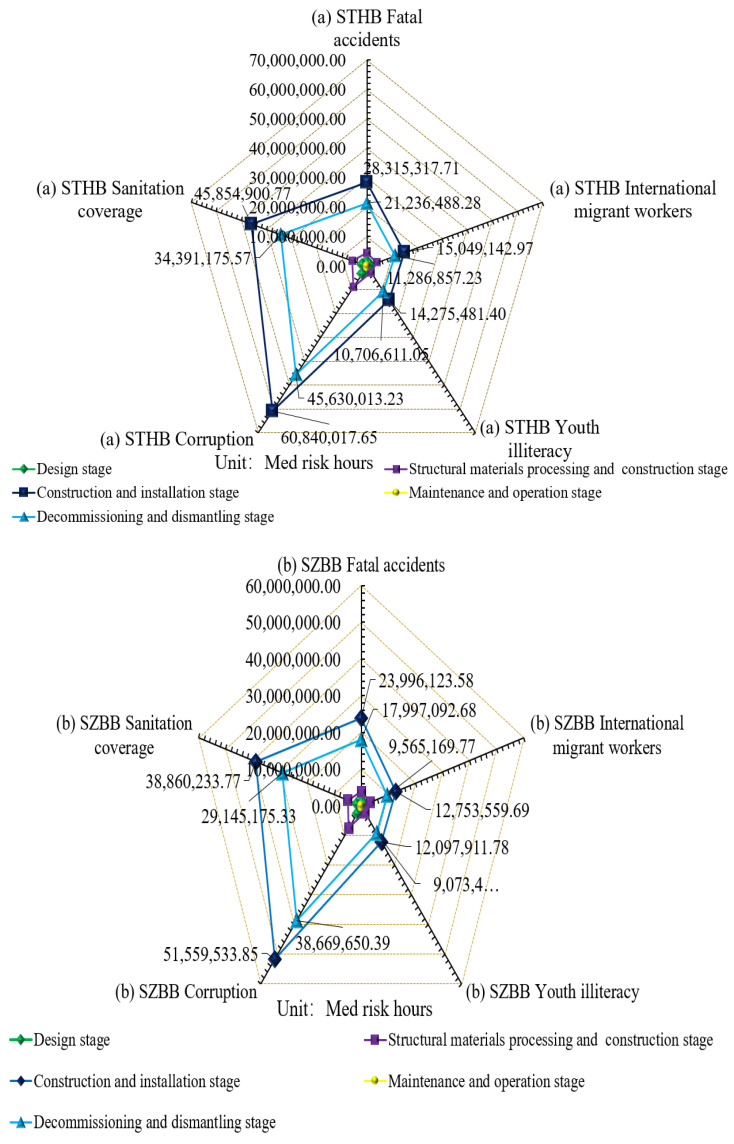
(**a**) The content in the first panel is the description of the five SILA factors of STHB; (**b**) The content in the second panel is the description of the five SILA factors of SZBB; (**c**) The content in the third panel is the description of the five SILA factors of BGNB; (**d**) The content in the fourth panel is the description of the five SILA factors of CJWB; (**e**) The content in the fifth panel is the description of the five SILA factors of XTHB; (**f**) The content in the sixth panel is the description of the five SILA factors of BSCB.

**Figure 9 ijerph-18-00122-f009:**
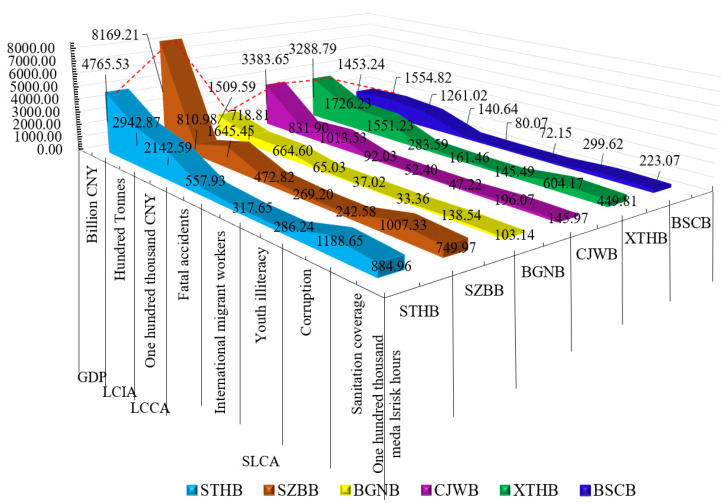
Schematic diagram of gross domestic product (GDP) [96], LCIA, LCCA, and SILA data in the area where the six cable-stayed bridges are located.

**Figure 10 ijerph-18-00122-f010:**
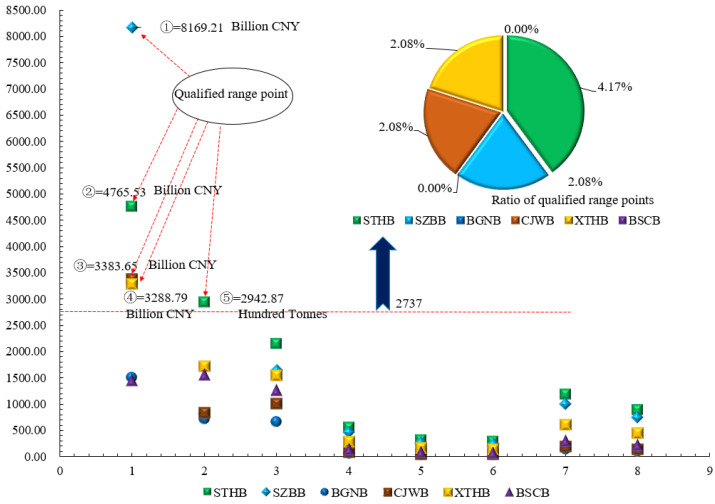
Schematic diagram of discrete points for comprehensive evaluation.

**Table 1 ijerph-18-00122-t001:** Recent statistics and analysis of some closely related achievements.

Methods	Description	Characteristic	Limitation	References
LCIA	Preventive design using 15 different methods of LCA concrete bridge deck.	How to reduce environmental pollution in the maintenance stage: Design and evaluation of 15 preventive measures.	The research content is relatively concentrated, single, and focuses on material replacement.	[16]
	Use LCIA to evaluate the rationality of the bridge design.	Use wooden bridges and alternative concrete to analyse the LCA impact of a cable-stayed bridge.	Ideal research design for the future. There are currently no large-span wooden bridges in operation. There are assumptions and uncertainties in the maintenance assessment of wooden bridges.	[17]
	Apply life cycle sustainability assessment to the superstructure of small span bridges.	The study was conducted using 27 bridges, and it was determined that a bridge composed of steel and concrete was the best indicator.	The LCA study of ordinary highway bridges, the conclusion is whether it is suitable for long-span special bridges.	[18]
	LCA was used to assess the environmental impact of the entire 60-year life span of the provincial highway.	The research structure has a complete range of tunnels, bridges, roadbeds, culverts, etc.	The road selection is in a remote area, and the research data are not representative.	[19]
	Several cases (schools, hospitals, commercial and residential buildings) were quantitatively studied using LCA.	There are many types of structures studied, and an evaluation model is established to quantitatively analyse emissions.	The research conclusions are poorly comparable, and the LCA data are highly uncertain.	[20]
LCCA	The article introduces a general framework for evaluating bridge life cycle performance and cost.	The focus is on analysis, prediction, optimization and decision-making under bridge uncertainty.	All the articles in this article are cost theory analysis, and there is no specific bridge case analysis.	[21]
	Research and develop an LCCA model to evaluate highway infrastructure investment.	Contributed to the systematic and informatised evaluation method of highway infrastructure investment.	Lack of case studies and model application research.	[22]
	The energy consumption cost of highway pavement is analysed based on LCCA and LCA.	Combining LCA and LCCA to determine the best pavement frame, road expansion projects are more practical.	Case application analysis of pavement concrete sustainability, no structural concrete evaluation.	[23]
	Quantify the life cycle environmental impact of the structure through environmental costs.	Calculate the environmental costs of materials, energy, transportation and construction equipment for the bridge structure.	The main research is the LCCA influence of the bridge girder structure.	[24]
	The LCC and LCA analysis of concrete bridges were discussed, and the optimization scheme was proposed.	Economic and environmental impact analysis of reinforced concrete and prestressed concrete bridges.	The bridge structure is simply a simply supported beam bridge across the river.	[25]
	Use SLCA to clarify the assessment (IA) methods and information covered in the different impact guidelines.	Use representational models to analyse the difference and connection between social influence and social performance.	All are written descriptions, without modeling and data analysis.	[26,27,28]
	Use SIA to study and practice all issues related to social issues in the entire project life cycle (before conception to after closure).	Analysed the overall social issues in the process of community and project management. Put forward that the biggest social problem management in the project is corruption.	Lack of case application analysis and discussion.	[29,30]
	SIA is undergoing a revolutionary force and revolutionary force for change.	SIA’s unfamiliarity with social sciences and the concerns of practitioners’ lack of competence.	Lack of case application analysis and discussion.	[31,32]
	EIA and SIA have technical flaws in analysis and evaluation.	Consider four conceptual elements in a sociological context of complexity and vitality.	Talked about the project SIA’s attention to sensitive factors and the improvement of social responsibility. How to realize the scientific methodology needs to be developed.	[33,34]
LCIA\LCCA\SILA	Evaluate the sustainability performance of different concrete and stone walls used in the building.	Multi-criteria decision analysis methods are used to evaluate and prioritise the alternative walls generated by LCA, LCC and S-LCA.	The research is sustainable and comprehensive, the evaluation structure is single, and recycling is not considered.	[35]
	The study analysed the impact of different mixed timber building structures on three different categories of environment, economy and society.	The comparison of wood and concrete in the building structure has been analysed to improve sustainability.	There are few studies on the three pillars of sustainability. This article has the same research route and different structures.	[36]
	Three box-type concrete bridges were optimised and sustainable.	Researchers focus on the environmental pillar, while the social pillar has been slow to develop.	It mainly studies the process of sustainability assessment and briefly analyses three precast concrete bridges.	[37]
	Discussed the framework for assessing the sustainability of bridges, including related technical, economic, environmental and social issues.	The sustainability of four versions of the same bridge was studied, and the local details of the bridge were analysed.	There is a lack of sustainable research on regional and actual operating bridges.	[38]

**Table 2 ijerph-18-00122-t002:** Cable-stayed bridge maintenance data statistics table.

Check Method	Inspection Cycle	Check Parts	Maintenance Cycle
Daily check	Working day	Pier foundation, cone slope, side wall of bridge abutment, pavement of bridge deck, drainage system, sidewalk, railing, guardrail, anti-collision wall of bridge deck, lighting system on bridge, expansion device, bridge head laying plate, sign, marking and traffic safety facilities, bridge installation sensors, wiring, cables, anchorage protection inspection, bridge damping device normal operation, support cleaning, rust and corrosion prevention.	Maintenance/year, Overhaul/5 years.
Frequency check	One time/every month		
Regular check	One time/one to three years	Coating layer of exposed concrete.	Maintenance/year, Replacement/5 years.
		Bridge deck paving, waterproof layer.	Maintenance/year, Overhaul/2 years, Replacement/10 years.
		Anti-collision guardrail, expansion joint.	Maintenance/year, Overhaul/2-5 years, Replacement/15 years.
		Cable-stayed bridge cables, slings, tie rods, external damping devices.	Maintenance/year, Overhaul/5 years, Replacement/20 years.
		Main beams, steel supports, bridge floor drainage pipes, bridge floor lighting facilities.	Maintenance/year, Overhaul/5 years, Replacement/50 years.
		Basin type rubber bearing.	Maintenance/year, Overhaul/5 years, Replacement/25 years.
		Damping device between towers and beams.	Maintenance/year, Overhaul/5 years, Replacement/30 years.
		Main beams, steel supports, bridge floor drainage pipes, bridge floor lighting facilities.	Maintenance/year, Overhaul/5 years, Replacement/50 years.

**Table 3 ijerph-18-00122-t003:** Cable-stayed bridge engineering data statistics table.

Bridge Name	Regional Location	Basic Situation	Bridge Layout Drawing
South Tai Hu Lake Bridge (338 m)	East China, Huzhou in Zhejiang	The main bridge is a double-cable, plane H-shaped, single-tower, concrete, cable-stayed bridge with a span layout of 160 + 190 + 38 m, an urban expressway level, and a design speed of 60 Km/h. The standard section width of the bridge is 40.5 m. The main beam adopts the cross-section form of double main ribs, the building height is 3.055 m, the full width is 40.5 m, and the standard main rib is 2.7 m high and 1.7 m wide. The transverse partition is 0.28 m wide; the bridge deck is 28 cm thick, and each cable plane has 24 pairs of cables.	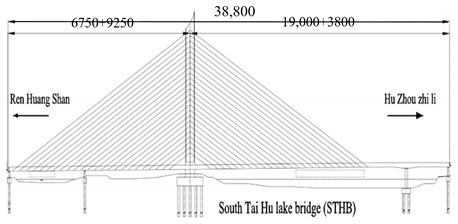
Shenzhen Bay Bridge (345 m)	Central and South China, Shenzhen Bay	The North Channel Bridge adopts the “180 + 90 + 75 m” span layout, the main beam adopts bolt-welded streamlined steel box girder, the beam height is 4.12 m, the standard section length is 12 m, and the overall width is 38.6 m. The total height of the pylon is 139.053 m. The main beam adopts a single-box, four-chamber, thin-walled structure composed of steel box beams with cantilever arms. The top plate thickness of the bridge deck is 18 mm; the bottom plate is 12–20 mm. The bridge has a total of 12 pairs of stay cables with a cable distance of 3 m and a standard cable distance of 12 m.	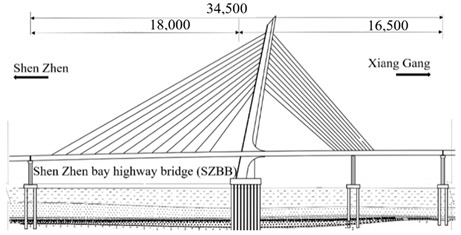
New Bridge of Xishuangbanna Tropical Botanical Garden (225 m)	Southwest China, Xishuangbanna Prefecture	The main bridge is an elliptical steel box with a concrete tower column, double cable plane, cable-stayed bridge with a span of 75 + 90 m and a total length of 165 m. The side span is 75 m and the main span is 90 m. The full width of the bridge deck is 14.2 m, the side main beam is 1.8 m high, the bottom width is 1.2 m, the outer top and bottom width is 1.55 m, and the bridge deck is 22 cm thick. The tower column of the cable-stayed bridge adopts a steel box concrete structure with a cross section of 2.5 * 4.0 m and a steel plate thickness of 20 mm.	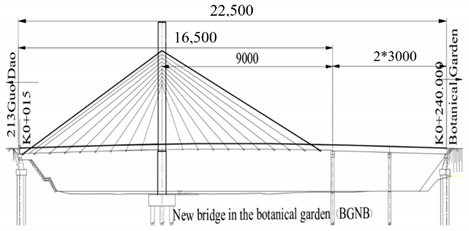
Cable-stayed Bridge of Changjiang West Road, Deyang City (136 m)	Southwest China, Deyang City	Single tower, single cable, plane cable-stayed bridge without back cable, main span 108 m, side span 27.7 m, harp-shaped cable surface, tower and beam consolidation. The standard cable distance on the beam is 8 m, the standard section is 8 m long and weighs about 300 Tons. The main beam adopts a pre-stressed concrete, single-chamber, three-box, flat, thin-walled box beam. The top plate of the box is 24 m wide; the bottom plate is 8.4 m wide, the beam height is 2.5 m, the top plate thickness is 24 cm, the bottom plate thickness is 30 cm, the inclined web plate thickness is 22 cm, and the vertical web plate thickness is 30 cm. A horizontal partition is set every 4 m with a thickness of 28 cm. The approach bridge adopts multi-span continuous beams, all of which are 20 m in span, and the main beam is a 1.4 m high box girder.	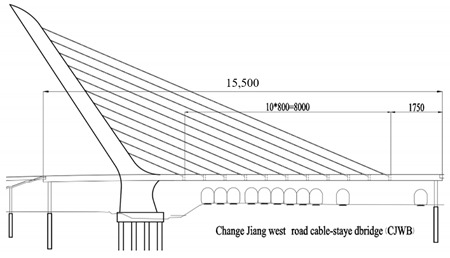
Hanjiang Highway Bridge in Xiantao City (312 m)	Central China, Xiantao City	The main bridge is a 50 + 82 + 180 m, three-span, single-tower, double-cable plane cable-stayed bridge, the main girder has a full cross-section width of 25.6 m, a basic section length of 8 m, a basic width of side ribs of 1.8 m, and a basic spacing of 8 m between the diaphragms. The roof thickness of the main beam is 0.30 m, and the beam height is 1.9 m.	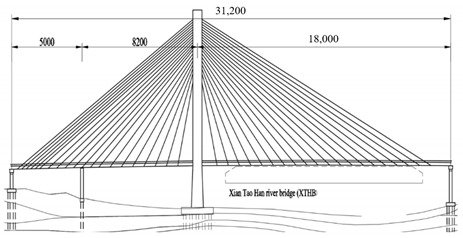
Baishan Bridge in Baishan City (410 m)	Northeast China, Baishan City	The main bridge is a two-span, single-cable, plane cable-stayed bridge with a span of 85 + 85 m. The main beam adopts a single box three-chamber section, the beam height is 2.0 m, the thickness of the top plate is 20 cm, and the thickness of the bottom plate is 40 cm. The section of the main tower adopts an “H” shaped cross-section concrete tower column. Oblique cable harp layout, single-cable deck bridge type, double-width layout with a net width of 15.5 m and a total width of 23.3 m.	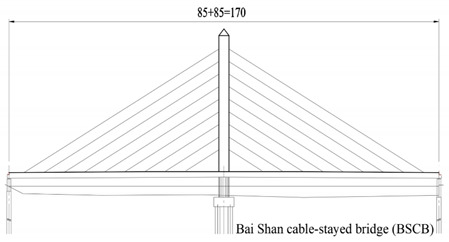

**Table 4 ijerph-18-00122-t004:** Statistical table of construction cost of six cable-stayed bridge projects ([91]). Unit: CNY.

Number	Cost Incurred	Ratio	Calculation Method	STHB	SZBB	BGNB	CJWB	XTHB	BSCB
1	Direct project cost			63,392,933.82	92,208,319.2	15,353,271.88	20,691,737.1	40,938,707.24	32,501,337.6
2	Insurance fee			1,901,788.015	2,766,249.576	460,598.1564	620,752.114	1,228,161.217	975,040.129
2-1	Project insurance stipulated in the contract	2.50%	1*2(2-1)	1,584,823.346	2,305,207.98	383,831.797	517,293.428	1,023,467.681	812,533.441
2-2	Third-party liability insurance stipulated in the contract	0.50%	1*2(2-2)	316,964.6691	461,041.596	76,766.3594	103,458.686	204,693.5362	162,506.688
3	Completion Files.	500,000	Constant cost	500,000	500,000	500,000	500,000	500,000	500,000
4	Construction environmental protection fees	1,000,000	Constant cost	1,000,000	1,000,000	1,000,000	1,000,000	1,000,000	1,000,000
5	Safety production fees	1.50%	1*5	950,894.0074	1,383,124.788	230,299.0782	310,376.057	614,080.6085	487,520.064
6	Engineering management software (temporary estimate)	100,000	Constant cost	100,000	100,000	100,000	100,000	100,000	100,000
7	Application fee for building information model technology	100,000	Constant cost	100,000	100,000	100,000	100,000	100,000	100,000
8	Temporary road construction, maintenance and dismantling fees			101,428.6941	147,533.3107	24,565.23501	33,106.7794	65,501.93158	52,002.1402
8-1	Fees for the construction, maintenance and dismantling of the original roads	0.08%	1*8(8-1)	50,714.34706	73,766.65536	12,282.6175	16,553.3897	32,750.96579	26,001.0701
8-2	Construction, maintenance and dismantling fees of temporary steel trestle and wharf	0.08%	1*8(8-2)	50,714.34706	73,766.65536	12,282.6175	16,553.3897	32,750.96579	26,001.0701
9	Temporarily occupying land and occupying the river	0.25%	1*9	158,482.3346	230,520.798	38,383.1797	51,729.3428	102,346.7681	81,253.3441
10	Erection, maintenance and dismantling of temporary power supply facilities	0.08%	1*10	50,714.34706	73,766.65536	12,282.6175	16,553.3897	32,750.96579	26,001.0701
11	Provision, maintenance and dismantling of telecommunications facilities	0.08%	1*11	50,714.34706	73,766.65536	12,282.6175	16,553.3897	32,750.96579	26,001.0701
12	Water supply and sewage facilities	0.08%	1*12	50,714.34706	73,766.65536	12,282.6175	16,553.3897	32,750.96579	26,001.0701
13	The construction fee of the contractor’s project department	0.42%	1*13	266,250.3221	387,274.9406	64,483.74189	86,905.2959	171,942.5704	136,505.618
14	Provisional expenses.	5.00%	(1 + 2 + 3 + 4 + 5 + 6 + 7 + 8 + 9 + 10 + 11 + 12 + 13)*14	3,431,196.012	4,952,216.129	895,422.4561	1,177,213.34	2,245,949.661	1,800,583.11
The total fees of the project			1 +…+ 14	72,055,116.25	103,996,538.7	18,803,871.58	24,721,480.2	47,164,942.89	37,812,245.2

**Table 5 ijerph-18-00122-t005:** Statistical table of loss from traffic accidents of six bridges during operation ([32]).

Bridge Name	Time Period (Years)	Accident Loss (CNY/Time)	Times of Accidents	Comprehensive Loss Fee (CNY)
STHB	2006~2018, 2019~2030, 2031~2105	3866	693\659\460	7,005,192
SZBB	2007~2018, 2019~2030, 2031~2106	3259	268\255\179	2,287,818
BGNB	2006~2018, 2019~2030, 2031~2105	4831	301\286\201	3,806,828
CJWB	2005~2018, 2019~2030, 2031~2104	8706	1070\1019\718	24,437,742
XTHB	2003~2018, 2019~2030, 2031~2102	6885	262\250\175	4,730,682
BSCB	2019~2030, 2031~2118	7213	456\434\306	8,626,748

**Table 6 ijerph-18-00122-t006:** Life cycle assessment (LCA) statistical tables for six cable-stayed bridges. Unit: kg.

Bridge Name	GWP	AP	FEP	PMFP	WP
STHB	285,792,121.03	758,359.05	778,387.38	2,755,862.99	4,202,670.97
SZBB	75,192,817.81	538,510.86	445,853.55	1,469,182.83	3,451,343.80
BGNB	69,261,736.42	214,170.43	251,077.34	756,768.56	1,397,595.57
CJWB	80,429,187.06	236,629.18	264,255.94	845,577.45	1,414,549.54
XTHB	167,606,486.66	424,005.32	502,313.61	1,559,831.83	2,530,246.34
BSCB	151,598,681.32	322,031.97	424,120.38	1,219,842.08	1,917,809.39

**Table 7 ijerph-18-00122-t007:** Environmental pollution data in Europe and China during the maintenance phase. Unit: kg.

Bridge Name	Quantity Analysed According to Chinese Standards	Quantity Analysed According to European Standards
STHB	202,577,714.70	4,060,953.15
SZBB	8,469,275.96	5,413,303.55
BGNB	46,427,579.22	1,264,900.09
CJWB	61,909,222.65	1,857,067.35
XTHB	127,556,952.20	3,689,371.79
BSCB	120,405,196.80	1,648,154.08

**Table 8 ijerph-18-00122-t008:** Statistical table of the cost ratio of 6 cable-stayed bridges.

Cost Name	STHB	SZBB	BGNB	CJWB	XTHB	BSCB
Cost of project preparation	0.01%	0.02%	0.01%	0.01%	0.01%	0.01%
Survey and design costs	0.07%	0.13%	0.06%	0.05%	0.06%	0.06%
Project construction costs	33.63%	63.20%	28.29%	24.39%	30.40%	29.99%
Maintenance and operating costs	60.57%	33.56%	63.69%	49.94%	63.78%	60.00%
Accident costs	3.27%	1.39%	5.73%	24.11%	3.05%	6.84%
Demolition stage costs	2.45%	1.69%	2.23%	1.50%	2.69%	3.10%

**Table 9 ijerph-18-00122-t009:** Statistical table of five social environmental impact data for 6 cable-stayed bridges. Unit: med risk hours.

Bridge Name	Fatal Accidents	International Migrant Workers	Youth Illiteracy	Corruption	Sanitation Coverage
STHB	55,792,892.84	31,765,165.76	28,624,476.33	118,864,998.3	88,496,114.86
SZBB	47,282,293.11	26,919,734.79	24,258,123.41	1,007,33434	74,996,993.87
BGNB	6,502,779.89	3,702,297.38	3,336,243.37	13,853,967.44	10,314,409.72
CJWB	9,202,951.4	5,239,614.97	4,721,563.11	19,606,597.66	14,597,297.3
XTHB	28,358,724.3	16,145,776.5	14,549,409.27	60,417,367.61	44,981,301.3
BSCB	14,063,615.15	8,006,988.77	7,215,320.78	29,962,088.48	22,307,058.1

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
