# Peer review of "Environmental, Economic and Social Impact Assessment: Study of Bridges in China’s Five Major Economic Regions"

_ijerph, 2020, doi:10.3390/ijerph18010122_

Round 1

Reviewer 1 Report

The article is interesting and deals with an important topic. It also fits in with the journal theme. The content does not raise any major reservations.
Minor Notes:
In the introduction, please describe similar research carried out in the world (even if they were only partial) - this will enable the comparison of the obtained results.
Improve the quality and readability Fig. 6.
There is no discussion of the obtained results - they could be based on the results presented in Fig. 9.
The conclusions are too general - they should relate to the results of the research presented in the paper

Author Response

All authors sincerely appreciate the comments and suggestions from Reviewer #1 about the paper. As regards improvements, we have tried to take into account the reviewer’s suggestions as discussed in the following point-by-point response. We have changed the manuscript accordingly.

Reviewer 2 Report

Thank you for the opportunity of reviewing your work.

The topic is very important and relevant, due to many implications of the issue. The research is well designed, and the methodology and results are well presented.

I have one concern related to the relatively weak section for literature and previous studies, which I consider that must be enhanced. More studies investigating the same topic by using similar / different methods should be listed and critically presented.

The conclusions must be enhanced too, and limitations and further research should be more extensively addressed, not only mentioning them. 

The abstract should be revised, it is very unclear now. 

A review of all paper must be performed, there are some errors/ typos.

Good luck!

Author Response

All authors sincerely appreciate the comments and suggestions from Reviewer #2 about the paper. As regards improvements, we have tried to take into account the reviewer’s suggestions as discussed in the following point-by-point response. We have changed the manuscript accordingly.

Reviewer 3 Report

The paper deals with the life cycle of bridges taking into account the environment, economy and society impact. The Authors present a study on six bridges located in different regional economic zones in China. Several key impact factors were defined through the definition of a weighted model using dedicated software.

Overall the paper is interesting and well-fit the purpose of the journal with results that are congruent with the hypotheses. However, the organization of the contents should be partially revised. In particular, in the current version of the manuscript the section "Method" is very complicated to follow and should be simplified.

I think the paper can be considered for the published in IJERPH journal once the section the section "Method" was reorganized. Also the English text editing should be revised.

Author Response

All authors sincerely appreciate the comments and suggestions from Reviewer #3 about the paper. As regards improvements, we have tried to take into account the reviewer’s suggestions as discussed in the following point-by-point response. We have changed the manuscript accordingly.

Reviewer 4 Report

Reviewed paper is certainly interesting, original and refers to vital and actual issues.

As Authors declare in the text, it fills the gap in the literature regarding environmental, economic and social impacts in China and its regions, in this case in light of large communication infrastructure.

For those reasons, I opt for acceptance of this paper for publishing.

There are however elements of the text that rise my questions and doubts.

My basic doubt rises in Point 2.1.1, where, in my opinion, the purpose is not clearly explained in this crucial chapter.

Another doubt refers to the prediction of data beyond 2020: Fig. 4 please elaborate on the methodology of estimation of the data on this Figure for years beyond 2020 – it is unclear and understanding of the methodology will be crucial to understand and review of the results.

It is also not clear why Authors chosen those bridges, as they are of different size, carrying systems/geometry, dominant structural material, location resulting radically different traffic and loads. Each of those bridges will, and it is clearly shown in the text, different impact due to listed above reasons. It is then hard to compare them and create continuous function out of discrete results. Certainly, as each of the cases is well described and analysed, each of them brings important data to understand global processes through the local situation. Formulation of continuous functions and general universal outcomes is then, in my opinion, unfounded (as i.e . Fig. 6 – right upper corner – such function has no sense, especially as the order of bridges taken to consideration is not objective – please change it into diagram with discrete results).

Another doubt refers to social impacts taken into consideration in Figure 9 – such problems as youth illiteracy or corruption are not explained sufficiently in the text and discussed in conclusions.

Moreover, I have objections to editorial aspects of the paper. There are numerous flows in:

Formatting:

Abstract – wrong text’s justification

Lack of spaces: v.14, v.38, v. 44, v. 47, v.53, v.57 – and more in the text.

Too large spaces between the words: v.14

Unnecessary spaces: v.51, 63, v. 276 – and more in the text.

Lack of punctuation marks: v.70

Too many of punctuation marks, e.g. v.50

Wrong punctuation marks e.g. v.37

Abbreviation for LCA – missing, should be in v.50

Abbreviations, sometimes normal text, sometimes italic (v.118)

Formulas – legends are difficult to read, perhaps each of symbol would be clearer explained in separate row.

Symbols should be written in italics also in legends.

Fig. 5 – the main beam of SZBB is a steel structure – unclear, perhaps whole carrying system is a steel structure

Fig. 6 – “Radio” – perhaps Ratio?

In general – too many formatting, editorial and grammar errors for the text to be published. Text has to be carefully checked and verified.

Author Response

All authors sincerely appreciate the comments and suggestions from Reviewer #4 about the paper. As regards improvements, we have tried to take into account the reviewer’s suggestions as discussed in the following point-by-point response. We have changed the manuscript accordingly.

Round 2

Reviewer 4 Report

Many thanks to Authors for exhausting explanations and extensive changes in the text. I assume, that text in present form can be published and will be certainly interesting to the readers.